# Biomarkers of Clot Activation and Degradation and Risk of Future Major Cardiovascular Events in Acute Exacerbation of COPD: A Cohort Sub-Study in a Randomized Trial Population

**DOI:** 10.3390/biomedicines10082011

**Published:** 2022-08-19

**Authors:** Peter Kamstrup, Jannie Marie Bülow Sand, Charlotte Suppli Ulrik, Julie Janner, Christian Philip Rønn, Sarah Rank Rønnow, Diana Julie Leeming, Sidse Graff Jensen, Torgny Wilcke, Alexander G. Mathioudakis, Marc Miravitlles, Therese Lapperre, Elisabeth Bendstrup, Ruth Frikke-Schmidt, Daniel D. Murray, Theis Itenov, Apostolos Bossios, Susanne Dam Nielsen, Jørgen Vestbo, Tor Biering-Sørensen, Morten Karsdal, Jens-Ulrik Jensen, Pradeesh Sivapalan

**Affiliations:** 1Section of Respiratory Medicine, Department of Medicine, Copenhagen University Hospital Herlev-Gentofte, 2900 Hellerup, Denmark; 2Nordic Bioscience A/S, 2730 Herlev, Denmark; 3Department of Respiratory Medicine, Copenhagen University Hospital Hvidovre, 2650 Hvidovre, Denmark; 4Department of Clinical Medicine, Faculty of Health and Medical Sciences, University of Copenhagen, 2200 Copenhagen, Denmark; 5The North West Lung Centre, Wythenshawe Hospital, Manchester University NHS Foundation Trust, Manchester M13 9PL, UK; 6Division of Infection, Immunity and Respiratory Medicine, University of Manchester, Manchester M13 9PL, UK; 7Pneumology Department, Hospital Universitari Vall d’Hebron, Vall d’Hebron Institut de Recerca (VHIR), Vall d’Hebron Barcelona Hospital Campus, 08035 Barcelona, Spain; 8Department of Respiratory Medicine, Copenhagen University Hospital Bispebjerg, 2400 Copenhagen, Denmark; 9Department of Pulmonary Medicine, Antwerp University Hospital, Laboratory of Experimental Medicine and Pediatrics, University of Antwerp, 2610 Antwerp, Belgium; 10Department Respiratory Disease and Allergy, Aarhus University Hospital, 8000 Aarhus, Denmark; 11Department of Clinical Medicine, Aarhus University, 8200 Aarhus, Denmark; 12Department of Clinical Biochemistry, Copenhagen University Hospital Rigshospitalet, 2100 Copenhagen, Denmark; 13Centre of Excellence for Health, Immunity and Infections (CHIP), Copenhagen University Hospital Rigshospitalet, 2100 Copenhagen, Denmark; 14Department of Respiratory Medicine and Allergy, Karolinska University Hospital Huddinge, 141 86 Stockholm, Sweden; 15Department of Medicine, Karolinska Institutet, 171 77 Stockholm, Sweden; 16Viro-Immunology Research Unit, Department of Infectious Diseases, Rigshospitalet, University of Copenhagen, 2100 Copenhagen, Denmark; 17Department of Cardiology, Copenhagen University Hospital Herlev-Gentofte, 2900 Hellerup, Denmark; 18Department of Biomedical Sciences, Faculty of Health and Medical Sciences, University of Copenhagen, 2200 Copenhagen, Denmark

**Keywords:** COPD exacerbation, major cardiovascular events, coagulation, biomarkers, von Willebrand factor, cross-linked fibrin degradation

## Abstract

Cardiovascular diseases are common in patients with chronic obstructive pulmonary disease (COPD). Clot formation and resolution secondary to systemic inflammation may be a part of the explanation. The aim was to determine whether biomarkers of clot formation (products of von Willebrand Factor formation and activation) and clot resolution (product of fibrin degeneration) during COPD exacerbation predicted major cardiovascular events (MACE). The cohort was based on clinical data and biobank plasma samples from a trial including patients admitted with an acute exacerbation of COPD (CORTICO-COP). Neo-epitope biomarkers of formation and the activation of von Willebrand factor (VWF-N and V-WFA, respectively) and cross-linked fibrin degradation (X-FIB) were assessed using ELISAs in EDTA plasma at the time of acute admission, and analyzed for time-to-first MACE within 36 months, using multivariable Cox proportional hazards models. In total, 299/318 participants had samples available for analysis. The risk of MACE for patients in the upper quartile of each biomarker versus the lower quartile was: X-FIB: HR 0.98 (95% CI 0.65–1.48), VWF-N: HR 1.56 (95% CI 1.07–2.27), and VWF-A: HR 0.78 (95% CI 0.52–1.16). Thus, in COPD patients with an acute exacerbation, VWF-N was associated with future MACE and warrants further studies in a larger population.

## 1. Introduction

Chronic obstructive pulmonary disease (COPD) affects 210 million people worldwide, and causes 1.9 million deaths annually [1]. Cardiovascular diseases (CVDs) are common among COPD patients, with a high prevalence of congestive heart failure, coronary heart disease, peripheral vascular disease, arrhythmia, stroke, and unspecified cardiovascular disease [2]. CVDs are associated with an increased rate of hospitalization and mortality in COPD patients [3,4].

COPD is a heterogenous disease and is often associated with persistent systemic inflammation [5]. Systemic inflammation induces increased stress on the endothelium, leading to varying degrees of endothelial damage and dysfunction [6].

Damage of the endothelium leads to a healing process, where von Willebrand Factor (vWF) is an initiator via exposure to extracellular matrix proteins through the damaged endothelium, which leads to local platelet adhesion [7]. Initially, when vWF is formed, an n-terminal pro-peptide is released as a bi-product. Antibodies for this pro-peptide are available, making it possible to assess the quantity of vWF formed (as the biomarker VWF-N) [8]. During its activation, vWF unfolds, which reveals a cleavage-site for degradation by the metalloprotease ADAMTS13 (a disintegrin and metalloprotease with ThromboSpondin motif repeats 13) [9,10]. When vWF is cleaved by ADAMTS13 following activation, a neo-epitope is released, where targeted antibodies allow for the quantification of vWF activation (as the biomarker VWF-A) [8].

Patients with COPD have increased vWF levels and relative serum activity [11]. vWF-antigen levels have been associated with first-time coronary heart disease [12], all-cause mortality [13], and stroke [14]. Additionally, low levels of ADAMTS13 activity are associated with all-cause and cardiovascular mortality [13], as well as stroke [15]. In patients with chronic heart failure, a high ADAMTS13/vWF-ratio is associated with a lower risk of clinical events [16]. The neo-epitope markers VWF-N and VWF-A were analyzed in plasma samples from the ECLIPSE trial, and revealed an association of VWF-N with the chronic condition of emphysema, and VWF-A with prior exacerbations [17]. Additionally, in a dichotomized form, higher values of both vWF products were associated with an increase in all-cause mortality [17].

Clotting is a complex interplay between clot formation and clot resolution, with both processes happening simultaneously. Further, fibrin is cross-linked as the clot matures and stabilizes [18]. As part of the ongoing clot resolution, cross-linked fibrin is degraded, releasing bi-products, including X-FIB, which is a neo-epitope of the plasmin-mediated degradation of cross-linked fibrin [19]. Thus, both vWF and the products of fibrin degradation may help describe an ongoing process of endovascular damage, clot formation, and clot resolution in patients with COPD.

Several non-interventional studies have indicated an increased risk of cardio- and vascular events among persons with an increased plasma level of Vwf [12,20].

The marker of degradation of cross-linked fibrin (X-FIB) has previously been shown to be related to emphysema and dyspnea, and predict mortality in stable COPD patients [19]. D-dimer, another marker of fibrin degradation, has been related to a prognosis in both healthy individuals and patients with COPD [12,21,22,23]. Similarly, fibrinogen, the precursor of fibrin, has been approved as a biomarker to enrich drug trials with endpoints of acute exacerbation or mortality [24].

The above-mentioned properties of both clot formation and resolution, with vWF as an initiator of clot formation and X-FIB as a marker of clot resolution, led to the hypotheses that high plasma levels of these neo-epitope biomarkers reflecting vWF formation (VWF-N) and activation (VWF-A), and of fibrin clot resolution (X-FIB) at the time of acute exacerbation of COPD (AECOPD) were associated with future MACE. Additionally, we aimed to describe the behavior of each biomarker, comparing acute with stable phase values.

## 2. Materials and Methods

### 2.1. Patients

The study population originated from the randomized controlled trial, CORTICO-COP, as described in detail in a previous publication [25]. Briefly, patients presenting with an AECOPD within 24 h of hospital admission between August 2016 and September 2018 were eligible for the trial. Blood samples (full blood and plasma) from 318 patients with COPD hospitalized for an acute exacerbation (severe or very severe COPD GOLD (Global initiative for chronic Obstructive Lung Disease) stage C/D and age ≥ 40) were collected at time of hospitalization, and at 30 days follow-up for all patients. Of these, 299 patients had blood sampled for biomarker analysis at baseline; of these, 203 had blood sampled after 30 days. Patients were recruited at three different Respiratory Departments in the Capital Region of Denmark (Copenhagen University Hospital Herlev & Gentofte, Copenhagen University Hospital Bispebjerg, and Copenhagen University Hospital Hvidovre). A follow-up visit was performed 30 days after the onset of an exacerbation. In the present study, only participants who had the biomarkers in question measured at baseline were included (299 (94%) participants), see Figure 1.

### 2.2. Measurements

The Spirometry and Medical Research Council (MRC) Dyspnea Scale was performed at baseline and at the 30 day follow-up. Venous blood samples were collected at hospital admission and at the 30 day follow-up via venipuncture into vacutainer EDTA-coated tubes. Plasma was obtained via centrifugation at 2000× *g* for 10 min at 4 °C within 30 min of blood collection. EDTA plasma samples were stored in aliquots at –80 °C within 1 h and until analysed. Neo-epitopes of vWF formation (VWF-N), vWF activation (VWF-A), and clot resolution (X-FIB) were assessed in EDTA plasma using specific enzyme-linked immunosorbent assays (ELISAs) (Nordic Bioscience A/S, Herlev, Denmark), as previously described [8,26]. In brief, all assays employed mouse monoclonal antibodies specific for a neo-epitope: VWF-N quantifies the N-terminal of the vWF pro-peptide released upon formation using competitive ELISA [8], VWF-A quantifies the ADAMTS13-mediated degradation fragment of vWF using competitive ELISA [8], and X-FIB quantifies the plasmin-mediated degradation of cross-linked fibrin using sandwich ELISA [26].

### 2.3. Confounding Factors

Data on possible confounding factors were collected from patients at baseline. Data on previous ischemic heart disease and heart failure were supplied with lifetime data from registries. We identified the following potential confounding factors: Sex, age, smoking status (current, former, or never smoked), C-reactive protein, forced expiratory volume in one second (FEV1, % of predicted), patient-reported use of inhaled corticosteroids, ischemic heart disease, heart failure, atrial fibrillation/flutter, hypertension, hypercholesterolemia, peripheral vascular diseases, and diabetes mellitus type 2.

### 2.4. Outcome

Registry-based follow-up on MACE was performed during the 36 month follow-up after hospital admission for an AECOPD. Hospital-related events and deaths were respectively obtained by linking the CORTICO-COP data to the Danish National Patient Registry and the Danish Civil Registration System. MACE was defined as either death or any admission to hospital with one of the International Classification of Disease codes as the primary diagnosis (see Appendix A).

### 2.5. Reporting

Reporting was carried out in accordance with the STROBE (STrengthening the Reporting of Observational studies in Epidemiology) guidelines [27]. Prior to the conduction of this sub-study, a protocol was published online [28].

### 2.6. Statistical Analyses

Assuming an annual event rate of the primary outcome of 2% and a hazard ratio (HR) of 3 over 36 months follow-up, corresponding to a MACE rate of 6 vs. 18%, 326 participants were required.

Descriptive statistics was performed on the baseline data at the time of AECOPD. A comparison of baseline data was performed with a Chi-squared test if the expected observations were above five. When the expected observations were five or below, Fischer’s exact test was applied. For continuous data, the normally distributed data were compared with a *t* test, and not normally distributed data were compared with Mann–Whitney Wilcoxon test. Patients who did not experience a new AECOPD within 30 days formed the “COPD-stable group” at the 30 day follow-up. Patients in the COPD-stable group who had biomarkers at both index admission and follow-up was used to investigate the change in each biomarker between the acute and stable phases with a Wilcoxon Signed-Rank test, and their correlation was examined with Spearman’s correlation coefficient.

On the biomarkers in acute phase: Each of the three biomarkers were dichotomized at the upper quartile, allowing for a comparison of “high value groups” (upper quartile) with “low value groups” (three lowest quartiles) in a multivariable Cox proportional hazards regression analysis adjusted for the previously mentioned potential confounding factors.

Missing data was subject to Substantive Model Compatible Fully Conditional Specification multiple imputation [29] (for details, see Appendix A). For missing data, where the missing completely at random/missing at random assumption could be doubted, both the best-case and worst-case imputations were applied (see Appendix A for further information).

Data management, descriptive statistics, and comparison of biomarkers were performed with Statistical Analysis Software 9.4 (SAS Institute, Cary, NC, USA). Multiple imputations and combinations of results were conducted in R 4.1.3 (R Foundation for Statistical Computing, Vienna, Austria) with the SMCFCS 1.6.1 and MITOOLS 2.4, packages, respectively. Crude Cox proportional hazards regression models were conducted using the SURVIVAL 3.3-1 package.

### 2.7. Model Control

Each biomarker was investigated for interaction with both known ischemic heart disease and known heart failure. Both best-case and worst-case imputations were performed as sensitivity analyses.

Proportional hazards assumption was tested as the interaction between exposure and time, and linearity was tested for the continuous covariates.

### 2.8. Explorative Analyses

Additional analyses were performed for the primary outcome, with the exclusion of all-cause mortality and with all-cause mortality as the sole outcome.

As an explorative analyses, each biomarker was investigated as a continuous variable. Further, each biomarker was investigated for an association with all parts of the composite outcome with a Chi-squared test (or Fischer’s exact test, if the expected observations were below five). When a significant association was identified, a multivariable Cox proportional hazards regression analysis was applied.

## 3. Results

Patient characteristics at baseline are shown in Table 1. In total, 299 (94%) patients had biomarkers assessed at baseline. For the assessment of biomarker stability between the acute and stable phases, 171 patients were stable at the 30 day follow-up, and had biomarkers accessible for comparison between baseline and the 30 day follow-up.

None of the biomarkers were normally distributed; see Figure 2 for characteristics of the biomarkers at baseline. Both vWF markers were generally unchanged from exacerbation to the stable phase (VWF-N: *p* = 0.41, VWF-A: *p* = 0.68), whereas there was a decrease in the stable phase for the clot resolution marker, X-FIB, with a median difference of 10.74 ng/mL (IQR: −60.36–17.99, *p* = 0.0038); Figure 3A–C. X-FIB and VWF-A had a strong correlation from the acute to stable phase, whereas VWF-N had a moderate correlation; Figure 3A–C.

In total, 149 (50%) of the eligible participants experienced MACE in the study period. Of these, seven experienced acute myocardial infarctions, and one, a stroke; 21 were admitted with heart failure and 120 died of all causes.

### 3.1. Primary Outcome

In the unadjusted Cox proportional hazards regression, no difference between high and low levels of acute-phase X-FIB (Hazard Ratio (HR): 1.10 (95% confidence interval (CI): 0.77–1.60), *p* = 0.60) nor VWF-A (HR: 0.85 (95% CI: 0.58–1.24), *p* = 0.41) on MACE was observed, while VWF-N was associated with MACE, HR: 1.41 (95% CI 1.00–2.00, *p* = 0.056).

Adjustments for known and suspected confounding factors did not change our findings: Neither X-FIB nor VWF-A predicted future MACE, X-FIB (HR 0.98 (95% CI 0.65–1.48), *p* = 0.93), and VWF-A: (HR 0.78 (95% CI 0.52–1.16), *p* = 0.21), whereas VWF-N (HR 1.56 (95% CI 1.07–2.27), *p* = 0.02) significantly predicted future MACE (Figure 4).

Neither the worst-case nor the best-case imputations altered the results (see Appendix A for complete results from all primary and secondary Cox proportional hazards regression models).

### 3.2. Secondary Outcome

None of the three biomarkers were associated with future MACE, when death was excluded from the composite outcome and included as a competing risk (X-FIB: HR 1.23 (95% CI 0.49–3.06, *p* = 0.66), VWF-N: HR 1.76 (95% CI 0.75–4.12, *p* = 0.19), and VWF-A: HR 0.66 (95% CI 0.24–1.81, *p* = 0.42).

### 3.3. Explorative Outcomes

Analyzing each biomarker as continuous variables did not alter the results. Each of the four composites of the main outcomes were tested against the dichotomized biomarkers. Of these, only VWF-A and heart failure had a significant association (*p* = 0.03); see Table 2. When using heart failure as an event in the multivariable Cox proportional hazards model (considering the other parts of the composite outcome as competing risks), VWF-A had a HR for heart failure of 0.16 (0.02–1.21, *p* = 0.08). No interaction between VWF-N and VWF-A was identified.

Post-hoc explorative-adjusted Cox proportional hazards regressions were performed on each biomarker using all-cause mortality as outcome. Of these, VWF-N showed a significant association with all-cause mortality (HR 1.52 (95% CI 1.02–2.29), *p* = 0.04).

## 4. Discussion

We found that high levels of VWF-N, a biomarker of vWF formation, were strongly associated with an increased risk of MACE. However, neither a marker of clot resolution (X-FIB) or vWF activation (VWF-A) was associated with MACE. Our results were robust through unadjusted and adjusted analyses.

The two clot activation factors were largely unchanged in blood levels from the time when patients were admitted with an AECOPD to the time when they reached a stable phase, one month later.

We found an association with a higher risk of MACE for patients with high VWF-N levels, which was completely attenuated when all-cause mortality was excluded from the primary outcome. This is consistent with previous studies, where high levels of vWF-antigen were linked to an increased risk of MACE [20]. Accordingly, in an explorative analysis solely using all-cause mortality, we found an association between VWF-N and all-cause mortality, which is line with a previous study, where VWF-N was associated with all-cause mortality [17]. Our findings on the formation of von Willebrand factor (VWF-N) are in line with the previous findings of von Willebrand factor levels, where increased levels have been associated with all-cause mortality, stroke, and first-time coronary heart disease [12,13,14]. The probable mechanisms of this are both the underlying endothelial dysfunction, which may increase von Willebrand factor levels, as well as the thrombogenicity of von Willebrand factor itself [7,9]. VWF-A was not associated with any of our prespecified outcomes, but showed a near-significant association with heart failure. Subsequent multivariable Cox proportional hazards regression showed a trend towards a lower risk of heart failure admissions in patients with high VWF-A levels. A protective effect of increased activation of a clotting factor is contra-intuitive; rather, it seems more likely that patients with high VWF-A levels in this case are patients with a higher activity of the metalloprotease ADAMTS13, since ADAMTS13 activity is decreased in patients with congestive heart failure [16].

Our neutral finding regarding fibrin degradation contradicts the sparse available data. In the ECLIPSE trial, X-FIB had a predictive value for two-year mortality, with an adjusted HR of 1.48 [19]. Of note, the ECLIPSE study recruited COPD patients in a stable state, whereas our study was based on patients acutely admitted with an exacerbation of COPD, which we showed resulted in higher values of X-FIB. D-dimer, which is also a marker of fibrin degradation similar to X-FIB, has previously been shown to predict 1-year mortality when sampled upon admission for AECOPD and all-cause mortality in a cohort of healthy persons [22,23]. Additionally, high D-dimer levels have been shown to predict coronary heart disease in persons without prior coronary heart disease [12]. Due to few ischemic events in our study, we did not have sufficient power to investigate this outcome.

This study has several strengths; mainly that it was a multicenter prospective study. The population was well-defined, consisting of severe COPD patients with an acute admission due to AECOPD. The study was Good Clinical Practice monitored, and had complete data on the outcome. Furthermore, blood samples were standardized and performed during baseline and follow-up at three hospitals. Finally, missing data were sparse, and no data were missing regarding biomarkers or regarding the explored outcomes (0% missing).

Despite the above-mentioned strengths, our study has some limitations: First, according to the calculated sample size, the study did not reach sufficient numbers, although it was close. This may have decreased the sensitivity to detect signals, or in contrast, it may have increased the risk of false findings. Second, a small proportion of patients did not have spirometry performed. We tried to compensate for this by performing multiple imputations and best-case/worst-case analyses; the results were the same from these approaches, and given that the vast majority of patients had spirometry, we do not believe this to have influenced the main analyses decisively. Lastly, the descriptive statistics on the change in biomarkers from the acute to stable phase may be influenced by patients not completing follow-up. However, this did not affect the primary analyses of this study.

There is a need to understand the mechanisms of life-threatening events such as MACE in COPD patients, both for identifying patients at risk of these events, and to gain knowledge for how to treat these patients in the future, to protect against such events. Thus, while the direct clinical use of the marker is currently limited, it has the possibility to aid in the yet limited classification of COPD phenotypes, and to identify patients who need interventions to avoid MACE. Further, it highlights the importance of von Willebrand factor in COPD patients during acute admissions.

In conclusion, in this study, where we investigated the biomarkers of clot formation and resolution among severe COPD (GOLD 3–4) patients admitted with AECOPD, we confirmed an association between VWF-N and future MACE from other studies. This may help to explain the mechanism of increased cardiovascular events among persons with COPD exacerbation.

## Figures and Tables

**Figure 1 biomedicines-10-02011-f001:**
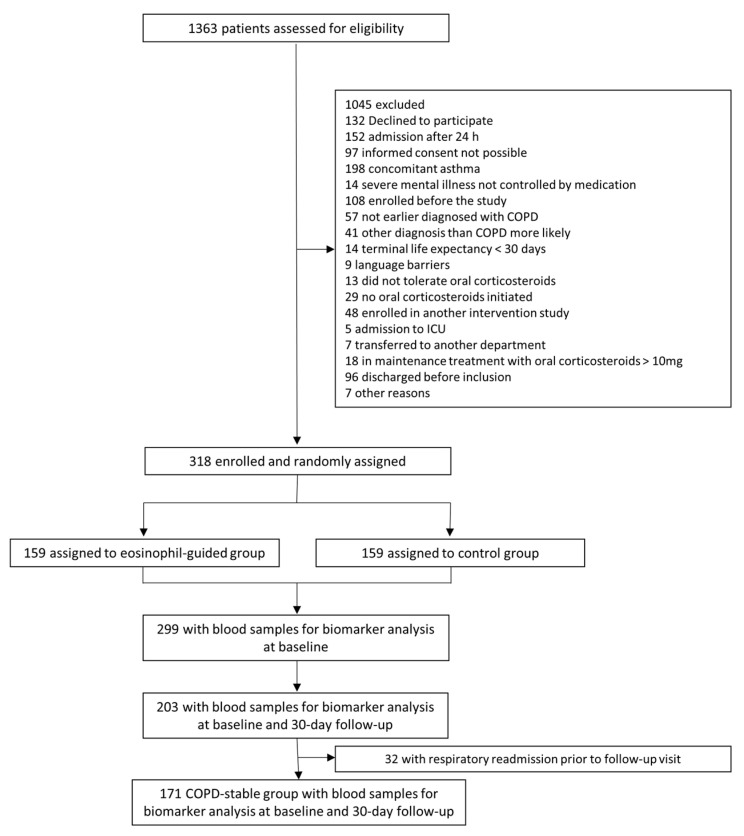
Study flowchart.

**Figure 2 biomedicines-10-02011-f002:**
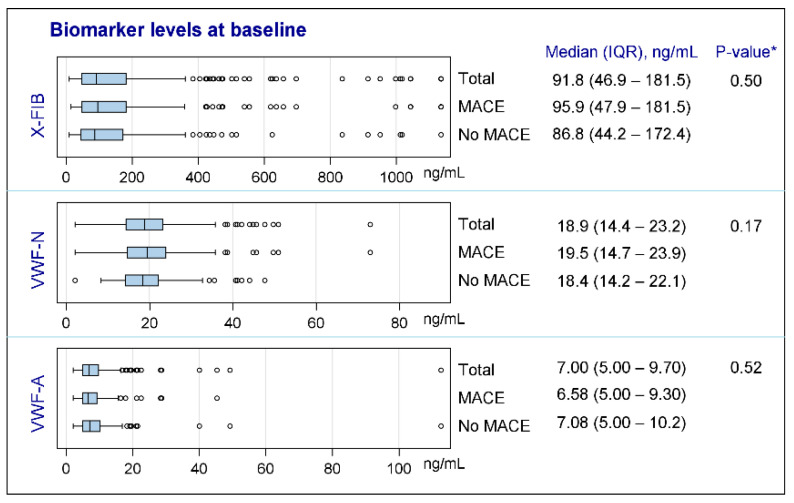
Characteristics of biomarkers at baseline. X-FIB: Cross-linked fibrin degeneration. VWF-N: N-terminal of von Willebrand Factor formation. VWF-A: Neoepitope of ADAMTS13-mediated activation of von Willebrand Factor. IQR: Interquartile range. MACE: Major cardiovascular event. *: For comparison between ‘MACE’ and ‘No MACE’ subgroups.

**Figure 3 biomedicines-10-02011-f003:**
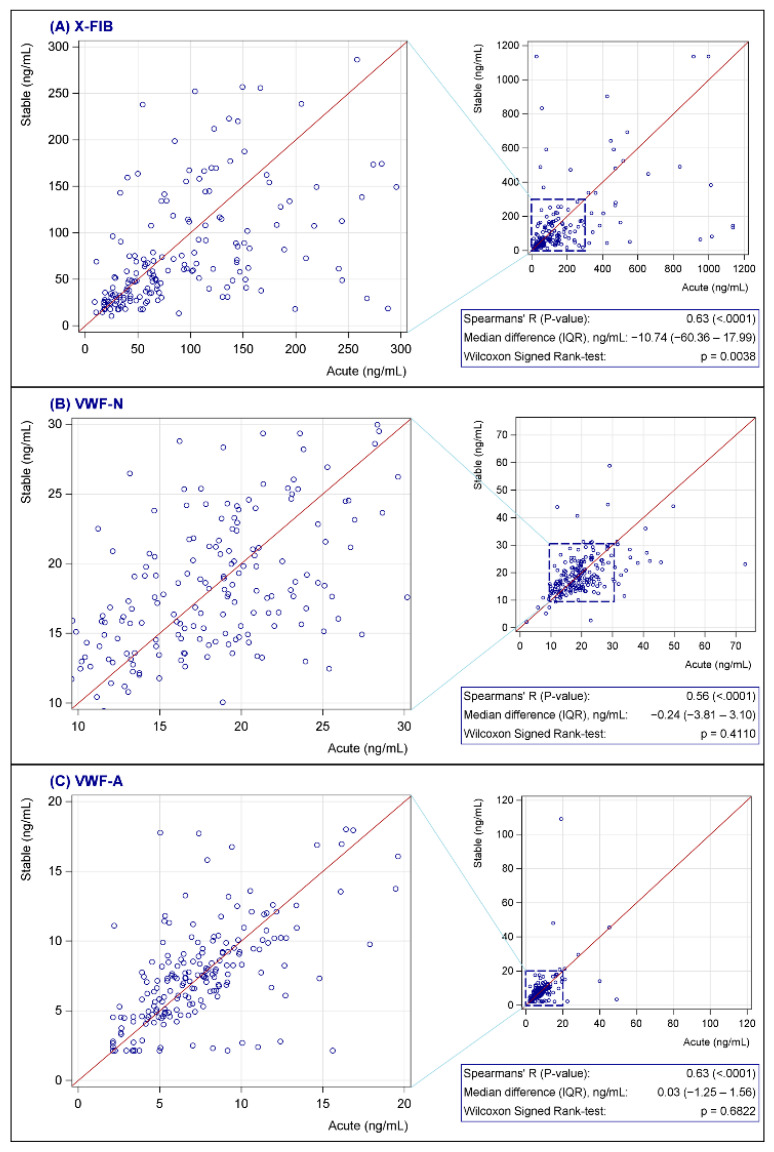
(**A**–**C**) Change in X-FIB (**A**), VWF-N (**B**), and VWF-A (**C**) from admission to follow-up. X-FIB: Cross-linked fibrin degeneration. VWF-N: N-terminal of von Willebrand Factor formation. VWF-A: Neoepitope of ADAMTS13-mediated activation of von Willebrand Factor. IQR: Interquartile range.

**Figure 4 biomedicines-10-02011-f004:**
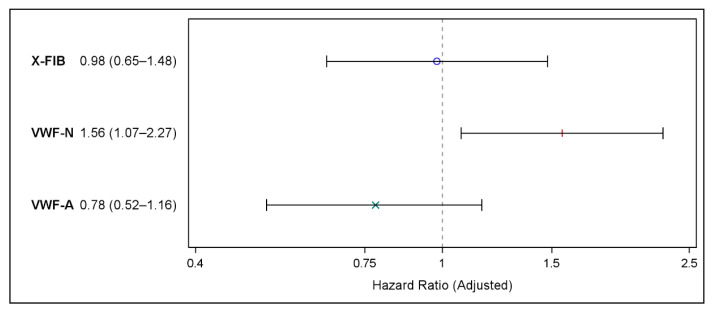
Forest plot showing the results of the multivariable Cox proportional hazards regression on risk of major cardiovascular events for X-FIB, VWF-N, and VWF-A; Hazard Ratio (95% confidence interval). Adjusted for age, sex, forced expiratory volume in one second (% of predicted), former ischemic heart disease, former heart failure, atrial fibrillation/flutter, hypertension, hypercholesterolemia, peripheral vascular disease type 2 diabetes, smoking status, and ICS use. °: X-FIB HR estimate, |: VWF-N HR estimate, ×: VWF-A HR estimate.

**Table 1 biomedicines-10-02011-t001:** Baseline characteristics. MACE: Major cardiovascular event, *n*: Number, SD: Standard deviation, IQR: Interquartile range, FEV_1_: Forced expiratory Volume in 1 s, MRC: Medical research council dyspnea scale score. *: For ever-smokers.

	Total (*n* = 299)	MACE during Study (*n* = 149)	No MACE during Study (*n* = 150)	Missing Data		*p*-Value
				MACE	No MACE	
Sex (male) sex, *n* (%)	141 (45)	68 (46)	67 (45)	0	0	0.87
Age (years), mean (SD)	75 (9)	77 (9))	73 (9)	0	0	<0.01
BMI (kg/m^2^), median (IQR)	24 (20–27)	23 (20–26)	25 (22–28)	1	2	<0.01
FEV_1_ (% of predicted), median (IQR)	31 (23–40)	29 (23–37)	34 (24–46)	9	10	<0.01
MRC, median (IQR)	4 (3–5)	4 (3–5)	4 (3–4)	2	3	<0.01
CRP (mg/L), median (IQR)	10.0 (26.0–86.0)	22.5 (11.0–68.0)	32.0 (7.0–97.0)	3	3	0.45
Smoking status						
Current smoker, *n* (%)	100 (33)	56 (19)	44 (15)	0	0	0.28
Former smoker, *n* (%)	194 (65)	91 (30)	103 (34)	0	0	
Never smoked, *n* (%)	5 (2)	<5 (<2)	<5 (<2)	0	0	
Pack years (years) *, median (IQR)	47 (30–57)	50 (30–60)	45 (30–56)	5	2	0.51
Medication						
Inhaled long-acting muscarinic antagonist, *n* (%)	234 (78)	120 (81)	114 (76)	0	0	0.34
Inhaled long-acting β2-agonist, *n* (%)	238 (80)	121 (79)	117 (81)	0	0	0.64
Inhaled corticosteroid use, *n* (%)	169 (55)	83 (56)	81 (54)	0	0	0.77
Long-term oral corticosteroid, *n* (%)	20 (7)	14 (9)	6 (4)	0	0	0.06
Comorbidities						
Previous MACE, *n* (%)	129 (43)	80 (54)	49 (33)	0	0	<0.01
Heart failure, *n* (%)	81 (27)	53 (36)	28 (19)	0	0	<0.01
Ischemic heart disease incl. stroke, *n* (%)	88 (29)	52 (35)	36 (24)	0	0	0.04
Peripheral vascular disease, *n* (%)	41 (14)	27 (18)	14 (9)	0	0	0.03
Type 2 diabetes mellitus, *n* (%)	38 (13)	24 (16)	14 (9)	0	0	0.08
Hypercholesterolemia, *n* (%)	37 (12)	17 (11)	20 (13)	0	0	0.61
Hypertension, *n* (%)	117 (39)	54 (36)	63 (42)	0	0	0.31
Kidney failure, *n* (%)	21 (7)	<21 (<11)	<5 (<2)	0	0	<0.01
Atrial fibrillation or flutter, *n* (%)	55 (18)	38 (26)	17 (11)	0	0	<0.01

**Table 2 biomedicines-10-02011-t002:** Distribution in outcome for high (upper quartile) versus low (lower three quartiles) levels of each biomarker.

Outcome	X-FIB			VWF-N			VWF-A		
	High	Low	*p*-Value	High	Low	*p*-Value	High	Low	*p*-Value
MACE, *n* (%)	38 (51)	111 (50)	0.87	45 (60)	104 (46)	0.04	35 (47)	114 (51)	0.53
Myocardial infarction, *n* (%)	0 (0)	7 (3)	0.19	2 (3)	5 (2)	1.00	3 (4)	4 (2)	0.37
Stroke, *n* (%)	0 (0)	1 (<1)	1.00	1 (1)	0 (0)	0.25	1 (1)	0 (0)	0.25
Worsening of heart failure, *n* (%)	8 (11)	13 (6)	0.19	6 (8)	15 (7)	0.79	1 (1)	20 (9)	0.03
Death, *n* (%)	30 (40)	90 (40)	1.00	36 (48)	84 (38)	0.13	30 (40)	90 (40)	1.00

MACE: Major cardiovascular event, *n*: Number, X-FIB: Cross-linked fibrin degradation, X-FIB: Cross-linked fibrin degradation. VWF-N: N-terminal of von Willebrand Factor formation, VWF-A: Neoepitope of ADAMTS13-mediated activation of von Willebrand Factor.

## Data Availability

Data collected for the CORTICO-COP trial, including individual participant data and a data dictionary defining each field in the set, will be made available to others in form of deidentified participant data. Informed consent forms will not be available according to Danish legislation. These data will become available from 1 January 2023 upon request from investigators. Registry-based data will not be made available, due to Danish legislation.

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
