# Peer review of "Biomarkers of Clot Activation and Degradation and Risk of Future Major Cardiovascular Events in Acute Exacerbation of COPD: A Cohort Sub-Study in a Randomized Trial Population"

_biomedicines, 2022, doi:10.3390/biomedicines10082011_

Round 1

Reviewer 1 Report

Table 1 presents data only for males? Were there any female patients in the cohort? 

Middle part of Table 1 (Biomarkers) should be presented in a figure

In discussion: Paragraph 1 has a conclusion. Which data supports that conclusion? Also, in the discussion indicate whether there was a postive or negative correlation, weak or strong. 

Clinically, how is this manuscript relevant? 

Author Response

Please see the attachment. In the attachment is a point-to-point response. The comments are listed C1, C2, C3, etc. Responses correspond to this and are listed e.g. R_C1, R_C2, R_C3 etc.

Reviewer 2 Report

this is an interesting study exploring the association of clotting related biomarkers during AECOPD with subsequent MACE at 36 m follow up

comments:

1. the conclusion that VWF-N is associated with subsequent MACE appears to based on marginal statistical significance. The unadjusted hazard ratio had p value > 0.05. Adjust HR said to be 0.05.  

2. Table 2 should include overall MACE with corresponding values included.

3. For death in table two, it appears that the high vWF-N group had lower mortality than the high group, which contradicts the text 

Author Response

(The authors gave the same response as above.)

Reviewer 3 Report

The current study confirmed an association between VWF-N and future MACE, but without significant. Concering MACE is the primary outcome, some vital confounder factors such as  peripheral arterial disease history, stain treatment were not included in a multivariable Cox proportional hazard regression analysis.

Author Response

(The authors gave the same response as above.)

Round 2

Reviewer 1 Report

V2 is a great version of the manuscript.

Reviewer 2 Report

Figure 4 needs labels for the two panels. I assume the top is unadjusted cox regression HR and bottom one is adjusted. also legend should stat the outcome is MACE

Reviewer 3 Report

The current study confirmed an association between VWF-N and future MACE, both in unadjusted and adjusted models. The discussion part is short and need more information, such as potential mechanisms and impact of public health or clinical practice.
